# communications
# earth & environment

# Lunar ejecta origin of near-Earth asteroid Kamo'oalewa is compatible with rare orbital pathways

Jose Daniel Castro-Cisneros [1✉], Renu Malhotra [2] & Aaron J. Rosengren[3]

Near-Earth asteroid, Kamo'oalewa (469219), is one of a small number of known quasi-satellites of Earth; it transitions between quasi-satellite and horseshoe orbital states on centennial timescales, maintaining this dynamics over megayears. The similarity of its reflectance spectrum to lunar silicates and its Earth-like orbit both suggest that it originated from the lunar surface. Here we carry out numerical simulations of the dynamical evolution of particles launched from different locations on the lunar surface with a range of ejection velocities in order to assess the hypothesis that Kamo'oalewa originated as a debris-fragment from a meteoroidal impact with the lunar surface. As these ejecta escape the Earth-Moon environment, they face a dynamical barrier for entry into Earth's co-orbital space. However, a small fraction of launch conditions yields outcomes that are compatible with Kamo'oalewa's orbit. The most favored conditions are launch velocities slightly above the escape velocity from the trailing lunar hemisphere.

[1] Department of Physics, The University of Arizona, Tucson 85721 AZ, USA. [2] Lunar and Planetary Laboratory, The University of Arizona, Tucson 85721 AZ, USA. [3] Mechanical and Aerospace Engineering, UC San Diego, La Jolla 92093 CA, USA. ✉email: jdcastrocisneros@arizona.edu

Small bodies in planetary systems can share the orbit of a massive planet in a long-term stable configuration by librating in the 1:1 mean-motion resonance[1]; such configurations are referred to as co-orbital motion. Examples of co-orbital arrangements are known for many Solar-System planets, the most ubiquitous being the large population of Trojan asteroids co-orbiting with Jupiter. In the context of the idealized circular, restricted three-body problem (CR3BP), there are three main types of co-orbital states: Trojan/tadpole (T), horseshoe (HS), and retrograde satellite/quasi-satellite (QS)[2]. The two cases of interest, horseshoe and quasi-satellite, are shown in Fig. 1a, which are distinguished by the center of oscillation of their longitudes relative to Earth, of 180° and 0°, respectively.

Unlike the long-term stable population of Trojan asteroids co-orbiting with Jupiter, most near-Earth asteroids (NEAs) have chaotic orbits with dynamical lifetimes much shorter than the age of the Solar System[3], and asteroids stably co-orbiting with the Earth on such timescales are uncommon. An assessment of Earth's co-orbital companions shows a total population of at least twenty-one objects, with two Trojan-type, six in the QS state, and thirteen undergoing HS motion; all of these objects are in their co-orbital states only temporarily, typically on less than decadal timescales[4–6]. The recently discovered quasi-satellite of the Earth, (469219) Kamoʻoalewa, is exceptional among the Earth's co-orbitals due to the longer-term persistence of its HS–QS transitions[7–10].

Kamoʻoalewa's diameter is estimated to be 46–58 m[11], and its orbital elements are listed in Table 1, in which we observe that, although its semi-major axis is very close to Earth's, its orbital eccentricity and inclination are not atypical of NEAs. Its ephemeris over a few centuries in the past and in the future, obtained from the Jet Propulsion Laboratory's (JPL) Horizons web service, shows that the transition from HS to its current QS state occurred nearly a century ago; an event that will reverse in about 300 years when Kamoʻoalewa will again pass into a HS orbit (Fig. 1c).

Long-term numerical simulations indicate that these transitions will recur over hundreds of thousands or even millions of years[9,10,12]. This can be contrasted with Earth's first-known recurrent quasi-satellite, asteroid 2002 AA$_{29}$, whose future predicted QS state will last only for a few decades[13,14]. Kamoʻoalewa's close proximity to Earth and its unknown dynamical origin make it a scientifically compelling candidate for a future space mission[15,16].

Several hypotheses have been proposed for the origin of Kamoʻoalewa[11,12]. Sharkey and colleagues measured its reflectance spectrum and found it to have an L-type profile resembling lunar silicates[11], inconsistent with typical NEAs. These authors also concluded that Kamoʻoalewa is unlikely to be an artificial remnant from an earlier lunar mission. Its modest inclination could be indicative of a temporarily captured NEA, as is speculated for other planetary co-orbitals[17]. Its orbital eccentricity, however, is atypical of such captured co-orbital states found in numerical simulations, which generally range between 0.3 (Venus crossing) and 0.6 (Mars crossing)[18]. The other proposed scenarios are that Kamoʻoalewa might have originated in the Earth-Moon system, either from a hitherto undiscovered quasi-stable population of Earth's Trojans or as a lunar ejecta from a meteoroidal impact[11].

These latter scenarios for the provenance of Kamo'oalewa are at variance with prevailing theoretical models of near-Earth objects[19,20] as these models assume only the main asteroid belt and comets as sources of NEAs. As a check, we employed the NEOMOD simulator[20] and found that the latest model of NEAs does not account for Kamo'oalewa-like orbits.

The focus of the present work is to examine the hypothesis that Kamoʻoalewa originated as lunar ejecta. We approach this by numerically simulating test particles (TPs) launched from the Moon's surface and following their subsequent orbital evolution. We use a physically plausible range of launching speeds and directions and four representative launch locations (Fig. 2). The dynamical evolution of lunar ejecta has been previously investigated with numerical simulations[21]. While those authors focused on determining whether lunar ejecta impact the Earth or Moon or escape into heliocentric orbits, our work focuses on determining

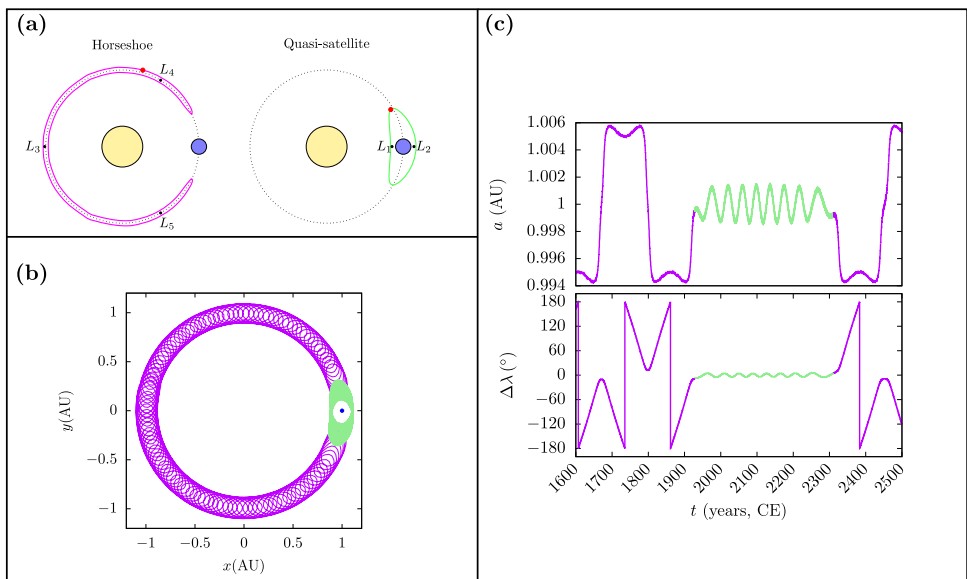

**Fig. 1 Co-orbital dynamics in the three-body problem and its relation to Kamo'oalewa's orbital dynamics. a** The two classes of co-orbitals considered in this work: horseshoe companions oscillate about the $L_3$ Lagrange point, diametrically opposite the planet's location, and encompass both $L_4$ and $L_5$ Lagrange points; and quasi-satellites orbit outside the planet's Hill sphere and enclose both the collinear $L_1$ and $L_2$ Lagrange points. **b** The trace of asteroid (469219) Kamo`oalewa's path in Cartesian coordinates in the co-rotating frame; Earth's position is shown in blue. **c** Kamo`oalewa's semi-major axis $a$ and relative mean longitude $\Delta\lambda = \lambda - \lambda_{\text{Earth}}$ as a function of time, with Horseshoe motion appearing in violet, while quasi-satellite motion is shown in green. The orbital propagation data for 1600–2500 CE are from JPL's Horizons ephemeris service (retrieved on 8 June 2023).

**Table 1 Orbital elements for Kamoʻoalewa.**

| Orbital elements | | Value | Uncertainty |
|---|---|---|---|
| Semi-major axis | $a$ (AU) | 0.9989754217067754 | $3.5408 \times 10^{-9}$ |
| Eccentricity | $e$ | 0.1064616822197207 | $2.4405 \times 10^{-7}$ |
| Inclination | $i(°)$ | 7.737141555926749 | $1.6932 \times 10^{-5}$ |
| Longitude of ascending node | $\Omega(°)$ | 67.69308146089658 | $1.4631 \times 10^{-5}$ |
| Argument of perihelion | $\omega(°)$ | 311.1680143115627 | $2.3126 \times 10^{-5}$ |
| Mean anomaly | $M(°)$ | 74.35927547581252 | $2.2534 \times 10^{-5}$ |

The orbit has been determined at epoch J2452996 (22 December 2003) from the JPL Horizons web-service (retrieved on 8 June 2023).

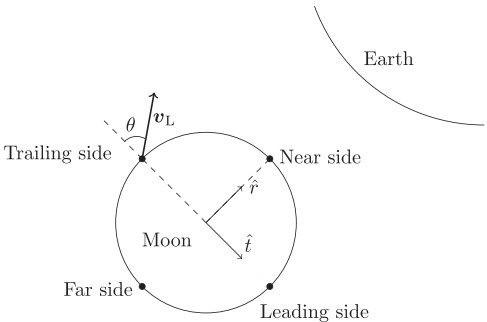

**Fig. 2 Initial conditions of lunar ejecta.** Launch conditions for test particles in terms of the parameters $\theta_1$, $\theta_2$, and $v_L$. The unit vector $\hat{r}$ defines the direction pointing towards Earth and $\hat{t}$ the transversal direction.

whether such particles have dynamical pathways that lead to co-orbital states. This is a more delicate question because, as we will see, such outcomes require statistically rare initial conditions (ICs); to our knowledge, this question has not been previously investigated.

## Results

As in previous numerical investigations of lunar ejecta, a variety of dynamical behaviors were found as particles entered heliocentric orbits. In order to depict the global results graphically, we projected the orbital evolution of the particles onto the semi-major axis–eccentricity plane, as shown in Fig. 3.

An immediately observable feature in this diagram is that most of the launched particles evolve into orbital parameter regions traditionally demarcated as the Aten and Apollo regimes of the population of near-Earth objects. A similar result has been reported previously[22]; it supports the suggestion that some of the members of the Aten and Apollo dynamical groups originate as lunar debris[12]. The other noteworthy feature in Fig. 3 is the vertical structure of a low density of points around $a = 1$ AU. This is the co-orbital region where we find the current orbit of Kamoʻoalewa and other HS–QS co-orbital NEAs. The evident well-defined boundaries of this region show that there is a dynamical barrier between lunar ejecta and co-orbital states but the finding of some outcomes in this region indicates that the barrier is somewhat porous, allowing a small fraction of lunar ejecta to evolve into and remain in co-orbital states for varying periods of time.

We identified the co-orbital outcomes by visual inspection of the time evolution of the particles that spend some time within the semi-major axis zone of 0.98–1.02 au. Overall, we found that 6.6% of all launched particles exhibited at least temporary co-orbital motion, most as HS (5.8%) and some as HS–QS (0.8%). A particle had to perform at least one HS or QS oscillation to be considered temporarily in a co-orbital state. A quantitative summary of the frequency for each dynamical outcome from each of the four launch sites is given in Table 2 along with the total

collisions detected from each site. The trailing side produced the most co-orbiters (both HS and QS), followed by leading side, and next by near side and far side which produced similar statistics.

Amongst all the initial conditions we simulated, some are more favorable for co-orbital outcomes than others. This is illustrated in Fig. 4; the histogram in Fig. 4b shows the distribution of the initial launch speed of the cases that resulted in HS and HS–QS outcomes. Overall, the most favored launch velocity for HS outcomes is near the minimum of the sampled range (i.e., just above lunar escape velocity); for HS–QS outcomes (i.e., Kamoʻoalewa-like) the most favored initial launch speed is moderately higher, ~(4.0–4.4) km s$^{-1}$, in agreement with the estimates from the Section "Theoretical estimates". In general, the total number of HS–QS outcomes decreases as the speed increases, discouraging exploration for larger values.

For additional detail, we examined the outcomes as a function of the radial and tangential components of the launch velocity (see Fig. 2 for an illustration of those directions), and made scatter plots of these velocity components for the four launch locations. In Fig. 4a, the initial launch velocity components from the lunar near-side are plotted as the yellow dots and those from the lunar far-side are in gray. In Fig. 4b, the initial launch velocity components from the lunar trailing-side are plotted as the yellow dots and those from the lunar leading-side are in gray. The HS and HS–QS outcomes are highlighted as the red and blue points. A clear asymmetry can be observed in these diagrams: most of the co-orbital outcomes were launched with a negative tangential velocity (i.e., in the trailing direction of the Moon's orbital motion). It is also evident that, out of the four representative launch sites considered, the trailing side is the most prolific in producing co-orbiters. Additionally it can be noticed that most of the co-orbitals produced from the leading side arise from the lower launch speeds, while for the other sites most of them arise from moderately larger launch speeds (>~3 km s$^{-1}$) (this will be shown to be due to the higher frequency of collisions for these conditions, as exposed in Fig. 5). We can also observe that for the larger launch speeds, in the range 4–6 km s$^{-1}$, co-orbital outcomes are favored for launch directions in the radial or anti-radial direction. These patterns in the outcomes are consistent with the theoretical expectations outlined in the Section "Theoretical estimates".

It is perhaps noteworthy that we did not find any tadpole-type outcomes, that is, particles librating around just the L4 or the L5 Lagrange points. Other possible fates that were examined were collisions. Collisions with the Moon and the Earth were registered, most of them occurring at the lower launch speeds and within the first 100 years of their evolution. The statistics of the collision outcomes is shown in Fig. 5, in a scheme analogous to Fig. 4. That is, panels (a) and (b) show the scatter plots of the initial conditions that end in collisions and panel (c) plots a histogram of the frequency of collisions at different launch speeds. We observe a clear, rapid decay of collision outcomes for larger launch speeds. The distribution appearing in Fig. 5a, b,

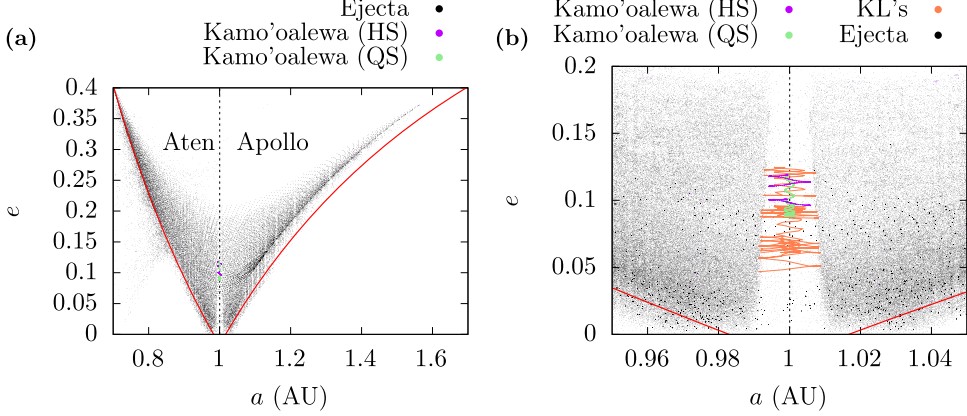

**Fig. 3 Summary of orbital outcomes of lunar ejecta particles in orbital parameter space. a** Orbital evolution for 5,000 years of 14,800 lunar ejecta particles, projected on the (*a, e*) plane. **b** The evolution of Kamo`oalewa and four different Kamo`oalewa-like (KL's) cases are highlighted on a zoomed-in portion of the diagram. The cadence of the non-co-orbital trajectories is downsampled (one point per 250 years) for legibility of the plots.

**Table 2 Summary of fates of lunar-ejecta TPs.**

| Launch site | # HS | # HS-QS | # Moon colliders | # Earth colliders |
|---|---|---|---|---|
| Near side | 189 (5.11%) | 24 (0.65%) | 31 (0.84%) | 301 (8.14%) |
| Trailing side | 281 (7.60%) | 42 (1.13%) | 41 (1.11%) | 373 (10.10%) |
| Far side | 157 (4.24%) | 24 (0.65%) | 36 (0.97%) | 303 (8.19%) |
| Leading side | 236 (6.37%) | 31 (0.84%) | 9 (0.24%) | 107 (2.89%) |

Frequencies of the orbital fates (including collisions with Earth and Moon) of the TPs from the four representative launching sites. Percentages of the total number of launched particles per site is shown in parenthesis. (Note that a TP may have reached a co-orbital state before colliding).

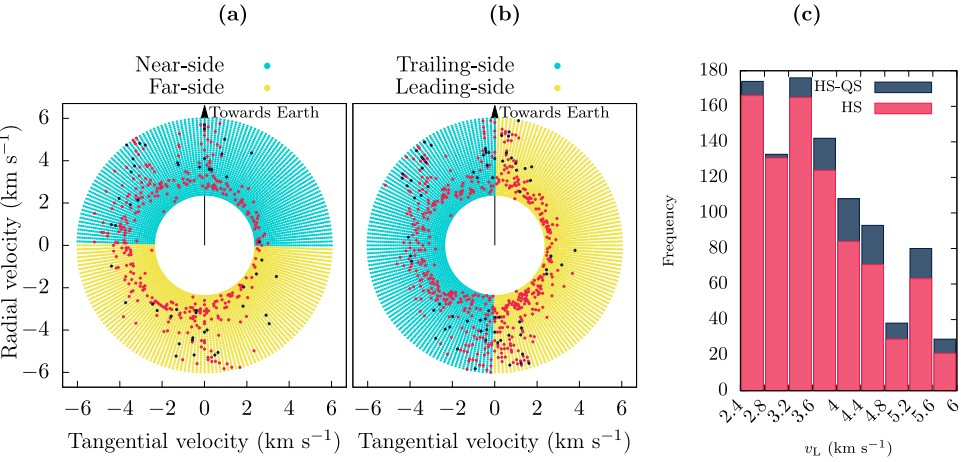

**Fig. 4 Outcomes of lunar ejecta particles related to their launch conditions. a, b** Each point represents a launch condition: one of four launch locations (near-side, far-side, leading side, and trailing side), and the launch velocity (decomposed into its radial and tangential components). See Fig. 2 for an illustration of the launch locations and launch velocity component directions. The points in red highlight those launch conditions that result in a HS state while the points in dark blue correspond to detected HS–QS transitions during the 5000 years of simulation time. **c** Histogram of the frequencies of co-orbital outputs with respect to the launching speed.

being concentrated at the lower speeds from near, far, and trailing side, accounts for the low frequency of co-orbital outcomes under these conditions, as a particle that may have reached such a state would have collided before it could enter into a co-orbital state.

Among the cases of HS-QS co-orbital outcomes observed in the simulations, most of them (around 66%) displayed only one transition or departed the QS state rapidly (before 1000 years), performing only one or two transitions. The orbits of interest are those whose HS–QS transitions recur persistently in a stable fashion for thousands of years, like Kamoʻoalewa. For the nine

ICs that showcased such Kamoʻoalewa-like dynamics (henceforth referred to as KL's; see Fig. 3), the evolution was tracked further, for up to 100,000 years, or until they departed their co-orbital states. Figure 6 shows an example KL outcome with persistent HS–QS transitions; this particle has recurrent residence times of 400–600 years in the QS state, in-between shorter residence times in the HS state. For comparison, Kamoʻoalewa's current time of residence in the QS state is ~400 years.

As previously noted, Kamoʻoalewa possesses a modest ecliptic inclination of about 8°. The orbital planes of most simulated

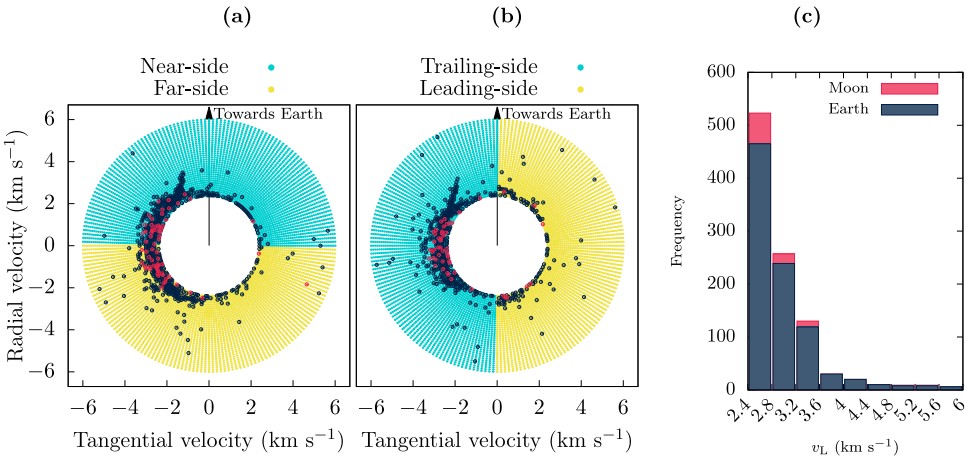

**Fig. 5 Collisional outcomes of lunar ejecta particles related to their launch conditions. a**, **b** Each point represents a launch condition: the radial and tangential components of the launch velocity and one of four representative locations on the lunar surface (near-side, far-side, trailing-side and leading-side). Points in red indicate collisions with the Moon, points in blue indicate collisions with Earth. **c** Histogram of the frequencies of collisions with respect to the launch speed.

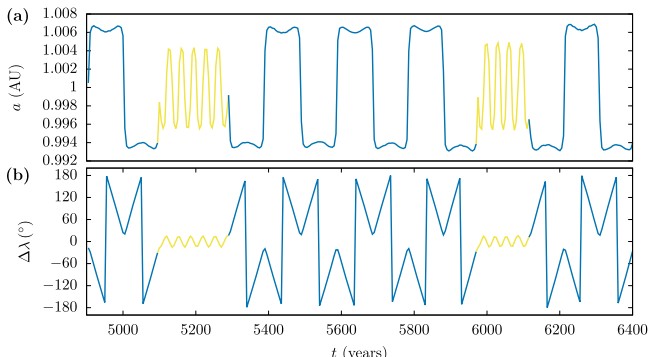

**Fig. 6 An example of a Kamoʻoalewa-like outcome of a lunar ejecta particle. a** Evolution of the semi-major axis and **b** relative mean longitude for the KL1 trajectory ($v_L = 5.1$ km s$^{-1}$). HS motion is shown in blue, while QS motion is shown in yellow. The HS-QS oscillations persist for up to 4500 years (not shown).

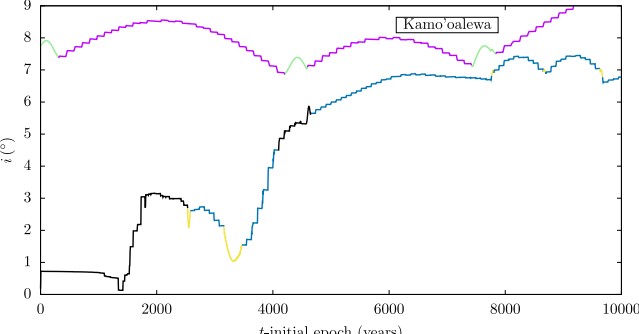

**Fig. 7 Evolution of the ecliptic inclination of Kamoʻoalewa and of a lunar ejecta particle.** For Kamoʻoalewa, HS states appear in violet and QS states in green. For the ejecta (KL2, $v_L = 4.4$ km s$^{-1}$), non-co-orbital motion is shown in black, HS states appear in blue and QS states in yellow. The orbital evolution was obtained with numerical propagation under the same model, starting at initial epoch J2452996.

ejecta particles were found to remain close to the ecliptic plane (typical inclinations ~ 1°–3°). This is because of our adopted simplification of considering initial launch conditions in a projected plane close to the ecliptic (see Fig. 2 and the associated description in the Section "Numerical model"). However, we did find some cases of KL outcomes in which the inclinations were driven up to higher values, reaching inclinations similar to Kamoʻoalewa's. This can be seen in Fig. 7, where we plot the time evolution of the inclination of a simulated particle (KL2), as well as that of Kamoʻoalewa's orbit. We observe episodic higher inclination states that persist for a few thousand years in-between short-lived low inclination states. The higher inclination states are reached by a sequence of inclination jumps that occur at close approaches between the particle and Earth, and these jumps build up coherently over time spans of a few hundred years. In this figure, we also plot Kamoʻoalewa's inclination evolution over a 10,000 year time span, revealing similar inclination jumps at close approaches with Earth during its HS state as well as similar features when transitioning from HS to QS states, but with a boost in inclination rather than a decrease. These results demonstrate that Kamoʻoalewa's inclination could have arisen from a smaller initial inclination by means of kicks at close approaches during its HS state.

## Discussion

It has been suggested that three small NEAs—2020 PN1, 2020 PP1, and 2020 KZ2, of estimated sizes in the range 10–50 m[23–25]—may have the same provenance as Kamoʻoalewa due to their close orbital clustering and the similarities they exhibit in their orbital evolutions on timescales of a few thousand years[5]. We have not investigated the orbital dynamics of these individual objects, but their resemblance to Kamoʻoalewa's orbital elements implies that our results for Kamoʻoalewa could also be applicable to these objects' origin.

The lunar ejecta hypothesis for the provenance of Kamoʻoalewa and other small Earth co-orbitals can be tested for consistency with the lunar impact crater record and cratering mechanics. The lunar ejecta velocities (in excess of lunar escape speed, 2.4 km s$^{-1}$) needed to obtain the co-orbital outcomes appear to be achievable in meteoroidal impacts on the Moon. Impacts on the Moon have typical impact speed of 22 km s$^{-1}$ and as high as 55 km s$^{-1}$[26,27]. Very small ejected debris particles may achieve comparable speeds, although the total fraction of such very high–velocity ejecta (solid or molten) is exceedingly small[28,29]. Based on studies of lunar secondary craters, it is estimated that an escaping lunar ejecta fragment of size in the tens of meters would be expected only from relatively large impact craters, of diameter

exceeding ~ 30 km[30,31]. During the past ~1100 Ma of lunar history (the Copernican period in the lunar geological timescale), there were 44 impact craters of diameter exceeding 30 km[32], indicating that such large impacts occur at average intervals of about 25 Myr. The implication is that if Kamoʻoalewa is a lunar impact ejecta fragment, then it was launched from the lunar surface $\mathcal{O}(10^7)$ years ago. We leave to a separate study to investigate whether a lunar crater of appropriate size and age and geographic location can be consistent with the lunar ejecta hypothesis for the provenance of Kamoʻoalewa. If supported by such studies, Kamoʻoalewa would, to the best of our knowledge, be the first near-Earth asteroid to be recognized as a fragment of the Moon. It would be of great interest for cosmochemical study as a sample of ancient lunar material. The rarity of Kamoʻoalewa-like orbital outcomes (compared to Aten- and Apollo-like outcomes) in our simulations of escaping lunar ejecta suggests that many other lunar ejecta remain to be identified amongst the background population of near-Earth asteroids. This prediction is testable with near-infrared reflectance spectra of very large numbers of NEAs that will be obtained by the forthcoming Near-Earth Object Surveyor project[33].

Additional exploration of the orbital evolution of lunar ejecta is also warranted. Our numerical investigations reported here were limited in a number of ways, so it is useful to list some future directions of investigation. In the present study, we identified the most-favorable launch velocities of lunar ejecta for Kamoʻoalewa-like outcomes for initial conditions of the Solar System taken near the present epoch. Although in the Section "Numerical model" we invoked the Copernican principle that the present epoch is not "special," we do recognize that our results may have some sensitivity to the initial epoch. The most important limitation is due to Earth's orbital eccentricity, which is time variable and undergoes excursions up to five times its current value on timescales of $\mathcal{O}(10^6)$ years. Consequently, Earth's orbital velocity varies by ~2 km s$^{-1}$, an amount that is comparable to (or a significant fraction of) the launch velocities of escaping lunar ejecta. Therefore, this could influence the frequency of the co-orbital outcomes of escaping lunar ejecta. Thus, sampling initial epochs when Earth's eccentricity is different is needed to understand more comprehensively the statistics of co-orbital outcomes of lunar ejecta. Sensitivity to epoch could also arise from lunar phase at launch (because the relative magnitude of solar perturbations on escaping lunar ejecta particles at full moon versus new moon phase is also a significant fraction of the lunar orbital velocity) and from the perturbations of Jupiter and other planets that would be slightly different at different epochs.

We would also like to understand the dynamical mechanism by which Kamoʻoalewa's persistent HS–QS transitions occur. Of the three possible co-orbital states, the QS state is the rarest found among small bodies in the Solar System. In the simple model of the planar, circular, restricted three body problem (PCR3BP), the intrinsic stability of nearly coplanar QS orbits has long been established[34–36]. It has been linked to the existence of the family $f$ of periodic orbits in the PCR3BP[37], and referred to as distant retrograde orbits (DROs) in applications to spacecraft navigation and mission design[38,39]. In the spatial problem, vertical instabilities can arise and transitions between co-orbital states are possible[40–44]. In the regime of large eccentricity and inclination[2], has attributed such transitions to a secular drift of the asteroid's perihelion and[12] has suggested this as a mechanism that applies to Kamoʻoalewa. While the eventual escape from co-orbital states may be linked to planetary secular perturbations[45–47], or to planetary close encounters[48], or to Yarkovsky-driven migration[10], it is likely that the short-time transport dynamics of Kamoʻoalewa are governed by the invariant-manifold structure of the Lagrange points[43,49,50]. For example, some authors attribute the entry and

escape mechanisms of Kamoʻoalewa's HS–QS transitions to such phase-space structures[50], but others invoke chaotic tangles of the Lagrange points to explain the dynamical mechanisms of capture into sticky QS orbits[38]. Nevertheless, it is challenging to identify the specific phase-space structures responsible for the dynamical transport phenomena exhibited by Kamoʻoalewa-like objects. More research is needed to understand the precise role of these manifolds on the dynamics of co-orbital objects like Kamoʻoalewa, as well as on the wider NEA populations, and their implications for the asteroid impact hazard on our home planet[44,51].

The complex and non-linear nature of the calculations performed leads to a large sensitivity to several conditions. For instance, initial conditions for objects in the Solar System were gathered from the JPL Horizons service, where masses and orbital elements are subject to updates and refinements. Further inconveniences arise from the fact that the orbital fates are classified based on visual inspection of 5000 year of orbital evolution of more than 10,000 simulated particles, so results are vulnerable to human error. Different results may also arise if the initial integration time of 5000 years is modified.

## Conclusions
In our numerical simulations of the dynamical fates of lunar ejecta, we explored a representative range of ejecta launch conditions expected from large meteoroid impact events. The vast majority (more than 93%) of the launch conditions we considered resulted in ejecta reaching heliocentric orbits similar to the Aten and Apollo groups of NEAs, with no co-orbital behavior detected; this is consistent with previous results[21,22]. However, in a small minority (6.6%) of cases we detected the existence of pathways leading to co-orbital states, most commonly horseshoe orbits, but also those resembling Kamoʻoalewa's; the latter exhibit persistent transitions between quasi-satellite and horseshoe orbits. These minority outcome events have not been previously reported. The existence of these outcomes lends credence to the hypothesis that Kamoʻoalewa could indeed be lunar ejecta. The launch conditions most favored for such an outcome are those with launch velocities slightly above the lunar escape velocity and launch locations from the Moon's trailing side. We also find that Kamoʻoalewa's inclination may have been boosted by close approaches with the Earth during its horseshoe state.

## Methodology
**Theoretical estimates**. We begin with the observation that particles originating in the Earth-Moon (EM) system that escape and evolve into Earth-like heliocentric orbits, including co-orbital states such as horseshoe and quasi-satellite types, would be those that escape with low relative velocity with respect to the EM barycenter. Here we make some estimates of the dynamical conditions of launch from the lunar surface that would favor outcomes with Earth-like heliocentric orbits.

For these estimates, we will take the Earth's Hill sphere as the approximate boundary between geocentric and heliocentric space, and the lunar Hill sphere as the approximate boundary between selenocentric and geocentric space. The radius of the lunar Hill sphere is ~35 lunar radii:

$$r_{H,\,\leftmoon} = \left[\frac{m_{\leftmoon}}{3\tilde{m}_{\oplus}}\right]^{\frac{1}{3}} a_{\leftmoon} \simeq 35.2\ R_{\leftmoon}, \qquad (1)$$

and Earth's Hill sphere radius is ~1% of Earth's heliocentric orbit radius:

$$r_{H,\oplus} = \left[\frac{\tilde{m}_{\oplus}}{3(m_{\odot} + \tilde{m}_{\oplus})}\right]^{\frac{1}{3}} a_{\oplus} \simeq 0.01\ au. \qquad (2)$$

Here $m_{\leftmoon}, \tilde{m}_{\oplus}, m_{\odot}$ are the lunar mass, the mass of Earth +

Moon, and the solar mass, respectively, $a_{\leftmoon}, a_{\oplus}$ are the lunar orbit radius and Earth's heliocentric orbit radius, respectively (both approximated as circular orbits), and $R_{\leftmoon}$ is the lunar radius.

We denote with $v_L$ the launch velocity of a particle launched from the Moon's surface; this is relative to the lunar barycenter. Particles launched with $v_L$ exceeding the Moon's escape speed, $v_{\leftmoon,esc} = 2.4$ km s$^{-1}$, will reach the lunar Hill sphere boundary with a residual speed $\delta v_L$ relative to the lunar barycenter. The magnitude of this residual velocity is estimated from the equation for conservation of energy in the lunar gravitational field, and is given by

$$(\delta v_L)^2 = v_L^2 - \frac{2Gm_{\leftmoon}}{R_{\leftmoon}}\left(1 - \frac{R_{\leftmoon}}{r_{H,\leftmoon}}\right) = v_L^2 - 0.97 v_{\leftmoon,esc}^2 \quad (3)$$

For later reference, we observe that for a particle launched with $v_L \approx v_{\leftmoon,esc}$, the residual velocity at the lunar Hill sphere radius is $\delta v_L \approx 0.4$ km s$^{-1}$.

The velocity of an escaping lunar ejecta particle relative to the Earth-Moon barycenter, $\boldsymbol{v}_{EM}$, is found by adding (vectorially) the residual velocity $\boldsymbol{\delta v}_L$ to the lunar orbital velocity, $v_{\leftmoon,orb}$,

$$v_{EM} = \delta v_L + v_{\leftmoon,orb} \quad (4)$$

The Moon's orbital velocity about the Earth-Moon barycenter is $v_{\leftmoon,orb} = 1.0$ km s$^{-1}$. To escape from the Earth-Moon system, the magnitude of $v_{EM}$ must exceed $\sqrt{2}v_{\leftmoon,orb} \simeq 1.4$ km s$^{-1}$. From Eq. (4), we see that this requires that $\delta v_L$ must exceed 0.4 km s$^{-1}$. This minimum value is coincidentally the same as the residual velocity at the lunar Hill sphere of lunar ejecta launched with just-the lunar escape speed, that is, $v_L \approx v_{\leftmoon,esc}$.

The magnitude of $\boldsymbol{v}_{EM}$ depends upon the location and speed of launch. We illustrate with two limiting cases. First consider lunar ejecta launched in the vertical direction from the apex of the lunar leading hemisphere. Such ejecta will reach the lunar Hill sphere boundary with residual velocity in a direction nearly parallel to the Moon's orbital velocity. Consequently, they will get boosted by ~1 km s$^{-1}$ (the lunar orbital velocity) to $v_{EM} > 1.4$ km s$^{-1}$, assuring escape from Earth's Hill sphere. The second limiting case is that of lunar ejecta launched vertically from the apex of the trailing lunar hemisphere. Such ejecta will reach the lunar Hill sphere boundary with residual velocity approximately anti-parallel to the Moon's orbital velocity, so their $v_{EM}$ will be lower than $\delta v_L$ by ~1 km s$^{-1}$. In this case a residual velocity of magnitude $\delta v_L > 2.4$ km s$^{-1}$ is needed in order to achieve $v_{EM} > 1.4$ km s$^{-1}$. From Eq. (3), this requirement implies a launch velocity $v_L > 3.4$ km s$^{-1}$. Ejecta from other locations and different launch directions will require a minimum launch speed in the range 2.4–3.4 km s$^{-1}$ in order to leave the Earth-Moon Hill sphere.

In the geocentric phase, the initial location of an escaping lunar ejecta particle is approximately at one lunar orbit distance, $a_{\leftmoon}$, and its velocity is $v_{EM}$ relative to the Earth-Moon barycenter. The ejecta will reach the Earth-Moon Hill sphere boundary with a residual velocity, $\delta v_{EM}$, relative to the Earth-Moon barycenter given by the energy conservation equation in geocentric space,

$$(\delta v_{EM})^2 = v_{EM}^2 - 2\frac{G\tilde{m}_{\oplus}}{a_{\leftmoon}}\left(1 - \frac{a_{\leftmoon}}{r_{H,\oplus}}\right) \simeq v_{EM}^2 - (1.2 \text{ km s}^{-1})^2. \quad (5)$$

Taking $v_{EM} = 1.4$ km s$^{-1}$ (the minimum required to achieve escape from geocentric space), escaping lunar ejecta will enter heliocentric space with a residual velocity (relative to the Earth-Moon barycenter) of $\delta v_{EM} = 0.7$ km s$^{-1}$. This is a small fraction, ~0.023, of Earth's heliocentric orbital velocity. Particles having $\delta v_{EM}$ close to this minimum value will enter heliocentric space with the most Earth-like orbits, and would be good candidates for

entering co-orbital states such as the horse-shoe or quasi-satellite orbits.

For high speed lunar ejecta, those launched in a direction opposite to the lunar orbital velocity achieve lower residual velocities relative to the Earth-Moon barycenter. This means that launch locations from the lunar trailing hemisphere (that is, the hemisphere opposite to the Moon's orbital motion around Earth) would be more favorable for Earth-co-orbital outcomes of escaping lunar ejecta. The circumstance that the Moon is in synchronous rotation with its orbital motion (and likely has been so for most of its history[52,53]) means that the favorable launch location can be geographically constrained in this way for even ancient epochs of launch times.

The above estimates are based on patching together three different point-mass, two-body models (Moon + TP, Earth + TP, and finally Sun + TP). For the purposes of these simple estimates, we have also ignored the effect of the Moon's rotation on the launch velocities of particles as well as the eccentricity of the lunar orbit and of Earth's heliocentric orbit. In detail, the orbital evolution of escaping lunar ejecta particles that enter Earth-like heliocentric orbits is subject to strongly chaotic dynamics and is exceedingly sensitive to initial conditions, as is well known in the three-body problem, hence the need for the numerical approach that follows below. These theoretical estimates provide a guide for the initial conditions of the lunar ejecta that are to be explored with numerical simulations and a guide for the analysis of the results.

**Numerical model**. We explore the dynamical fates of lunar ejecta in a similar vein as has been done for satellite ejecta in the Saturnian system[54]. Our dynamical model includes the eight major planets from Mercury to Neptune and the Moon, and we use the IAS15 integrator within REBOUND[55]. The Direct predefined module in REBOUND was used to detect collisions with the massive bodies. An initial step size of 1.2 days was used and the step–size control parameter was set to its default value ($\epsilon = 10^{-9}$; this assures machine precision for long time orbit propagations of $10^{10}$ orbital periods[55]). The length of the main set of simulations was 5000 years; this is sufficiently long to explore the details of possible co-orbital outcomes as a first study of the proof-of-concept for the lunar-ejecta hypothesis for the origin of Kamoʻoalewa. In a second set of simulations, we extended the simulation time up to 100,000 years for those particles exhibiting Kamoʻoalewa-like dynamical behavior. Running these simulations for much larger time spans is computationally prohibitive because, in order to detect the QS and HS dynamical states and transitions between such co-orbital configurations, it is necessary to have high cadence outputs (~1 output per year); this places high demands on data storage.

The initial conditions for the planets are obtained from the JPL Horizons system at epoch J2452996, i.e., 22 December 2003. In this initial exploration, we adopt the Copernican principle[56,57] that the current time is not special, and is not unrepresentative of lunar ejecta launch conditions at any random time in the geologically recent past. This assumption can be tested in the future by exploring different initial epochs that sample different initial conditions of the planets—especially of Earth's eccentricity —on secular timescales.

The initial conditions for the test particles are generated through three parameters: the angle $\theta_1$ between the line joining the center of the Moon to the launch site on the lunar surface and the line joining the center of the Moon to center of Earth, the angle $\theta_2$ between the launch velocity vector and the local normal vector at the launch site on the lunar surface, and the speed of launch $v_L$. For simplicity, we consider a two dimensional

projection of the Moon's surface onto the ecliptic, illustrated in Fig. 2. Accordingly, the position of the TP is completely specified by $\theta_1$ (and the Moon's radius), while its initial velocity (relative to the lunar surface) is determined by the magnitude and direction of the specified relative velocity ($v_L$ and $\theta_2$, respectively); according to the projection made, this relative velocity has no component perpendicular to the ecliptic plane. The angles $\theta_1$ and $\theta_2$ range from $(0°, 360°)$ and $(-90°, 90°)$, respectively. The values of $v_L$ were chosen between 2.4 and 6.0 km s$^{-1}$, with the lower bound corresponding to the lunar-escape speed and the upper bound being the limit of ejection velocities reported in numerical simulation studies of lunar cratering events[58]. It should also be noted that the frequency of ejection velocities decreases rapidly as $v_L$ increases[58], discouraging the exploration of larger values of $v_L$.

Following the guidance from the Section "Theoretical estimates", four launch sites were sampled on the Moon's surface. These four sites are representative of each of the four hemispheres of the Moon: the near-side, far-side, leading side, and trailing side. The locations of these are shown in Fig. 2. At each location, the launch speed was varied systematically from 2.4 to 6.0 km s$^{-1}$ (in increments of 0.1 km s$^{-1}$), and, for each speed, 100 particles were launched with different angles $\theta_2$ (uniformly chosen along $-90°$ and $90°$). In total, we launched 14,800 test particles.

In the simulations, we monitored for collisions with all the massive bodies. In order to identify co-orbital outcomes, we visually examined the time series of $a$ and $\Delta\lambda$. Rather than examining the time series of all launched particles, this task is made easier by first projecting the evolution in the $(a, e)$ plane and identifying those particles that appear in a rather sparsely populated, narrow vertical zone in the semi-major axis range $0.98 - 1.02$ au, as explained further in the "Results" section.

## Data availability
Outcomes of the numerical simulations presented in this paper are available at this publicly accessible permanent repository: https://doi.org/10.5281/zenodo.8339513.

## Code availability
REBOUND can be freely downloaded from the developers' webpage https://rebound.readthedocs.io. Codes and scripts used for the numerical simulations may be requested to the corresponding author.

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

## Acknowledgements

We acknowledge an allocation of computer time from the UA Research Computing High Performance Computing (HPC) at the University of Arizona. The citations in this paper have made use of NASA's Astrophysics Data System Bibliographic Services. R.M. additionally acknowledges research support from the program "Alien Earths" (supported by the National Aeronautics and Space Administration under agreement No. 80NSSC21K0593) for NASA's Nexus for Exoplanet System Science (NExSS) research coordination network sponsored by NASA's Science Mission Directorate, and from the Marshall Foundation of Tucson, AZ. A.R. acknowledges support by the Air Force Office of Scientific Research (AFOSR) under Grant No. FA9550-21-1-0191.

## Author contributions

J.D.C.C. wrote the codes and scripts used for the numerical simulations, conducted the simulations, data processing, and data analysis. R.M. composed the theoretical estimates and carried out related dynamical calculations. R.M. and A.J.R. designed the research project. All authors discussed and critiqued interim and final results and contributed to writing the manuscript.

## Competing interests

The authors declare no competing interests.
