## [Peer Review File · Communications Earth & Environment]

6th Jun 23

Dear Mr Castro-Cisneros,

Your manuscript titled "Orbital pathways for a Lunar-Ejecta Origin of the Near-Earth Asteroid Kamo`oalewa" has now been seen by 3 reviewers, and I include their comments at the end of this message. They find your work of interest, but some important points are raised. We are interested in the possibility of publishing your study in Communications Earth & Environment, but would like to consider your responses to these concerns and assess a revised manuscript before we make a final decision on publication.

We therefore invite you to revise and resubmit your manuscript, along with a point-by-point response that takes into account the points raised. Please highlight all changes in the manuscript text file.

Please use the following link to submit your revised manuscript, point-by-point response to the referees' comments (which should be in a separate document to any cover letter) and the completed checklist:

[link redacted]

We hope to receive your revised paper within six weeks; please let us know if you aren't able to submit it within this time so that we can discuss how best to proceed. If we don't hear from you, and the revision process takes significantly longer, we may close your file. In this event, we will still be happy to reconsider your paper at a later date, as long as nothing similar has been accepted for publication at Communications Earth & Environment or published elsewhere in the meantime.

We understand that due to the current global situation, the time required for revision may be longer than usual. We would appreciate it if you could keep us informed about an estimated timescale for resubmission, to facilitate our planning. Of course, if you are unable to estimate, we are happy to accommodate necessary extensions nevertheless.

Please do not hesitate to contact me if you have any questions or would like to discuss these revisions further. We look forward to seeing the revised manuscript and thank you for the opportunity to review your work.

Best regards,

Joe Aslin

Senior Editor,
Communications Earth & Environment

<https://www.nature.com/commsenv/>

Twitter: @CommsEarth

EDITORIAL POLICIES AND FORMATTING

Editorial Policy: [Policy requirements](https://www.nature.com/documents/nr-editorial-policy-checklist.pdf) (Download the link to your computer as a PDF.)

Furthermore, please align your manuscript with our format requirements, which are summarized on the following checklist:

[Communications Earth & Environment formatting checklist](https://www.nature.com/documents/commsj-phys-style-formatting-checklist-article.pdf)

and also in our style and formatting guide [Communications Earth & Environment formatting guide](https://www.nature.com/documents/commsj-phys-style-formatting-guide-accept.pdf) .

***** DATA:** Communications Earth & Environment endorses the principles of the Enabling FAIR data project (<http://www.copdess.org/enabling-fair-data-project/>). We ask authors to make the data that support their conclusions available in permanent, publically accessible data repositories. (Please contact the editor if you are unable to make your data available).

All Communications Earth & Environment manuscripts must include a section titled "Data Availability" at the end of the Methods section or main text (if no Methods). More information on this policy, is available at <http://www.nature.com/authors/policies/data/data-availability-statements-data-citations.pdf>.

If a community resource is unavailable, data can be submitted to generalist repositories such as

<https://figshare.com/>>figshare or Dryad Digital Repository. Please provide a unique identifier for the data (for example a DOI or a permanent URL) in the data availability statement, if possible. If the repository does not provide identifiers, we encourage authors to supply the search terms that will return the data. For data that have been obtained from publically available sources, please provide a URL and the specific data product name in the data availability statement. Data with a DOI should be further cited in the methods reference section.

Please refer to our data policies at http://www.nature.com/authors/policies/availability.html.

REVIEWER COMMENTS:

Reviewer #1 (Remarks to the Author):

Journal: Nature Communications Earth & Environment

Manuscript ID: NCOMMSENV-23-0603

Title: Orbital pathways for a Lunar-Ejecta Origin of the Near-Earth Asteroid Kamo'oailewa

Authors: J. D. Castro-Cisneros, R. Malhotra, and A. J. Rosengren

The authors of this manuscript investigate the plausibility of Lunar ejecta reaching "stable" orbits within Earth's co-orbital zone. This research was motivated by observational results presented in:

Lunar-like silicate material forms the Earth quasi-satellite (469219) 2016 HO3 Kamo'oailewa
<https://ui.adsabs.harvard.edu/abs/2021ComEE...2..231S/abstract>

The authors of this manuscript posted a preprint of the work under review here on April 27:

Orbital pathways for a Lunar-Ejecta Origin of the Near-Earth Asteroid Kamo'oailewa
<https://arxiv.org/abs/2304.14136>

Preliminary results of this research have already been presented elsewhere:

Earth's quasi-satellite Kamo'oailewa's possible origin as lunar ejecta
<https://ui.adsabs.harvard.edu/abs/2022cosp...44..189C/abstract>

Earth's Quasi-satellite Kamo'oailewa's Possible Origin as Lunar Ejecta

<https://ui.adsabs.harvard.edu/abs/2022DDA....5320404C/abstract>

Near-Earth Asteroid Kamo`oalewa as Lunar Ejecta

<https://ui.adsabs.harvard.edu/abs/2021DDA....5230503C/abstract>

This subject has been researched before as shown in some of the works cited by the manuscript under review here, but this is the first time the topic is analyzed within the context of Earth's co-orbital small body population. The manuscript arrives to the conclusion that a small but significant percentage of simulated Lunar ejecta may end up temporarily trapped in orbits not too different from that of Kamo`oalewa. The results presented in the manuscript are based on a careful and detailed analysis of the dynamical fates of lunar ejecta resulting from a representative and plausible range of ejecta launch conditions. They are indeed of great interest for the entire scientific community.

For the reasons pointed out above, I am glad to recommend this manuscript for publication in Nature Communications Earth & Environment after somewhat minor revision. This minor revision request is motivated by the fact that the authors did not include an analysis on how frequent are present-day small scale lunar impacts that may inject meteoroids in these orbits, but also that Lunar material may be present in the main asteroid belt and elsewhere. On the other hand, considering the results presented here within the context of popular NEO models may help in deciding which dynamical pathway could be more fitting in this case. I encourage the authors to explore and discuss these issues in a future revised version of their manuscript.

Minor issues

=====

1) Please, use the latest orbit determination of Kamo`oalewa released by JPL's Small-Body Database:

https://ssd.jpl.nasa.gov/tools/sbdb_lookup.html#/?sstr=469219

2) How likely is to find objects with orbital parameters close to those of Kamo`oalewa within the framework of the NEOPOP model? See:

<https://neo.ssa.esa.int/neo-population-generator>

Kamo`oalewa may be somewhat contemporary Lunar ejecta but also it might have appeared long ago when Lunar maria and major craters were formed and then be ejected towards the main asteroid belt or beyond. Then it may have returned to the neighborhood of Earth in a fashion somewhat similar to what may have happened to Manx comets:

Inner solar system material discovered in the Oort cloud
<https://ui.adsabs.harvard.edu/abs/2016SciA....2E0038M/abstract>

3) Frequency of small scale Lunar impacts issue. The authors have constrained the analysis of the frequency of formation of Lunar impact craters to craters with a diameter of tens of km. The authors argue that based on studies of Lunar secondary craters, it is estimated that an escaping Lunar ejecta fragment of size in the tens of meters would be expected only from relatively large impact craters, of diameter exceeding 30 km. It is true that Kamo`oalewa has tens of meters in diameter but the authors also mention three other smaller minor bodies, namely 2020 PN1, 2020 PP1, and 2020 KZ2 that may have the same origin and may have been produced in smaller scale impact crater formation events. Some relevant references from the literature could be:

Lunar impact flash results and space surveillance activities at Kryoneri Observatory
<https://ui.adsabs.harvard.edu/abs/2023arXiv230300670L/abstract>

Present-day model of lunar meteoroids and their impact flashes for LUMIO mission
<https://ui.adsabs.harvard.edu/abs/2023Icar..38915180M/abstract>

A new chronology from debiased crater densities: Implications for the origin and evolution of lunar impactors
<https://ui.adsabs.harvard.edu/abs/2023E%26PSL.60217963X/abstract>

Topographic Diffusion Revisited: Small Crater Lifetime on the Moon and Implications for Volatile Exploration
<https://ui.adsabs.harvard.edu/abs/2022JGRE..12707510F/abstract>

Reviewer #2 (Remarks to the Author):

There is a simple equation in Melosh (2020) allowing to estimate the size of the largest fragment ejected from an impact crater. It may be better to use it (and the result does not contradict your estimates). It is obvious from this equation that the fragment size correlates negatively with an impact velocity - there is no need to mention exceptionally high impact velocities on the Moon.

The statement "ejected debris may achieve speeds up to 6 km/s" (line 546) is not quite correct - the speed could be as high as an impact velocity, but the fraction of this super-high speed ejecta is negligible. Also it is worth to mention that the total fraction of solid high-velocity ejecta is relatively small in comparison with molten high-velocity ejecta.

The total number of >30-km-diameter craters of Copernican period could be estimated from the Neukum's production function (also a very easy exercise) with the reference to Neukum et al. (2001) paper. It is indeed ~44.

Reviewer #3 (Remarks to the Author):

This paper explores the hypothesis that Kamo was originated from the fragments from a meteoroidal impact with the lunar surface, with intensive numerical simulations. The results are exciting and supports this hypothesis to a very good level. It is a very interesting and original research to the community and the wider field. Some main concerns include: 1) how is the reliability of the numerical simulations, like how to validate your numerical simulations, maybe even partially; 2) could you please discuss the extent of the sensitivity? For instance, if the initial condition is perturbed for about 1%, how much the error of the orbit will be at 5000 years time? In addition, does the theoretical assumption have an influence on the numerical conclusion?

The statistical analysis is appropriate but its validity needs to be discussed more. I think the level of detail provided are good enough to reproduce the work

Some minor comments given here:

Page 6 line 242, Please justify why the accuracy is set to be $10e-9$, rather than other values such as $10e-12$?

Page 6 line 254

Why you select 22th Dec 2003 for the initial conditions? Please explain what is Copernican principle? Please add reference

Page 6 line 260, please explain the definition of θ_1 and θ_2

Fig.7 Is the inclination evolution of Kamo from pure numerical simulations? Are there any observation data that can fit part of the evolution curve?

Manuscript COMMSENV-23-0603

Response to Reviewers

Authors' responses are in bold font.

REVIEWER 1

This minor revision request is motivated by the fact that the authors did not include an analysis on how frequent are present-day small scale lunar impacts that may inject meteoroids in these orbits, but also that Lunar material may be present in the main asteroid belt and elsewhere. On the other hand, considering the results presented here within the context of popular NEO models may help in deciding which dynamical pathway could be more fitting in this case. I encourage the authors to explore and discuss these issues in a future revised version of their manuscript.

Minor issues

1) Please, use the latest orbit determination of Kamo'oailewa released by JPL's Small-Body Database: <https://ssd.jpl.nasa.gov/#/?sstr=469219>

Table 1 has been updated from JPL's database (retrieved on 8 June 2023).

2) How likely is to find objects with orbital parameters close to those of Kamo'oailewa within the framework of the NEOPOP model? See: <https://neo.ssa.esa.int/neo-population-generator>

Based on a published realization of this synthetic NEO population generator (Granvik et al., 2018), we found zero objects with orbital parameters among the values reached by Kamo'oailewa along the 900 years provided by JPL Horizons. This is illustrated below in Fig. R1 which plots the 900-year track of Kamo'oailewa in (a, e, i) parameter space, and there are no model NEOs in this parameter range.

Kamo'oailewa may be somewhat contemporary Lunar ejecta but also it might have appeared long ago when Lunar maria and major craters were formed and then be ejected towards the main asteroid belt or beyond. Then it may have returned to the neighborhood of Earth in a fashion somewhat similar to what may have happened to Manx comets: Inner solar system material discovered in the Oort cloud <https://ui.adsabs.harvard.edu/abs/2016SciA....2E0038M/abstract>

The reviewer sketches an alternate hypothesis that an ancient lunar ejecta from several gigayears ago could have taken an elaborate circuitous route of traveling to very large heliocentric distances, residing there for several gigayears, then [somehow] returning to co-orbit with the Earth. The reviewer points to such a scenario proposed by Meech et al 2016 for the origin of the Manx comets*. The sequence of necessary steps in such a scenario for Kamo'oailewa would be as follows: launch from Moon into near-Earth heliocentric orbit, get further scattered, get deposited in a long term stable orbit in the main asteroid belt or beyond, survive there for several gigayears, then return to a co-orbital state with Earth. While we cannot rule out this scenario in the N-body Solar system, we consider it highly implausible for the following reasons. In the first two steps, the largest orbits that scatterings by Earth can lead to are nearly parabolic orbits with perihelion near Earth but their aphelion cannot reach more than about 2 au, i.e., near the inner edge of the main asteroid belt. (At this stage, only hyperbolic orbits reach heliocentric distances larger than 2 au, in which case they escape from the solar system and cannot return.) Therefore ejecta can hardly reach the "main asteroid belt or beyond" without additional aid. For the third step: Close encounters with Mars and/or asteroids or the distant perturbations of the giant planets would be weak (and not analogous to the stellar and Galactic perturbations for the Manx comets); they would not lead to stable orbits within the

main asteroid belt or beyond. The last step of entering a co-orbital state with Earth from a near-parabolic return orbit of large semimajor axis is energetically prohibitive (though it can be achieved with careful rocket engineering for spacecraft). For example, an orbit of semimajor axis 2.5 au and perihelion distance of 1 au would need to dump 60% of its orbital energy and about 25% of its orbital angular momentum to transfer to an Earth-like orbit, unlikely with natural (including non-gravitational) forces acting on rocky asteroids. Moreover, depositing the object into the delicate dynamical niche of a quasi-satellite of Earth would require exquisite fine-tuning of the actions of those putative forces. A future investigation could assess the (exceedingly low) probability of this scenario.

*For the Manx comets, the scenario proposed by Meech et al 2016 is that rocky asteroidal material originally formed at heliocentric distances of 3.5–13 au was scattered into near-parabolic orbits by Jupiter and/or Saturn, then followed the path of Oort Cloud comets, namely evolving with a slow diffusion of their aphelion and of their perihelion (under stellar and Galactic perturbations) into distant near-circular orbits residing in the Oort Cloud for several gigayears, then being perturbed (also by stellar and Galactic perturbations) into near-parabolic comet-like orbits that make brief incursions into the inner solar system. (Note that the initial conditions in this scenario are in the outer asteroid belt, not near Earth; for the reasons explained in the previous paragraph, near-Earth origin of Manx comets is implausible.) The initial orbits would be under the influence of the giant planets (much more massive and more distant planets compared to Earth), and can therefore be scattered to orbits of large aphelion distances where they come under the influence of the Galaxy. This also means that the return orbits are of large semimajor axis, near-parabolic, not near-circular. The Manx comets’ near-parabolic orbits are not known (nor theorized) to return to near-circular orbits in the inner solar system, and certainly not into Kamo‘oalewa-like co-orbiting with Earth.

3) Frequency of small scale Lunar impacts issue. The authors have constrained the analysis of the frequency of formation of Lunar impact craters to craters with a diameter of tens of km. The authors argue

Fig. R1

Fig. R2

that based on studies of Lunar secondary craters, it is estimated that an escaping Lunar ejecta fragment of size in the tens of meters would be expected only from relatively large impact craters, of diameter exceeding 30 km. It is true that Kamo'oailewa has tens of meters in diameter but the authors also mention three other smaller minor bodies, namely 2020 PN1, 2020 PP1, and 2020 KZ2 that may have the same origin and may have been produced in smaller scale impact crater formation events. Some relevant references from the literature could be:

Lunar impact flash results and space surveillance activities at Kryoneri Observatory <https://ui.adsabs.harvard.edu/abs/2023arXiv230300670L/abstract>

Present-day model of lunar meteoroids and their impact flashes for LUMIO mission <https://ui.adsabs.harvard.edu/abs/2023Icar..38915180M/abstract>

A new chronology from debiased crater densities: Implications for the origin and evolution of lunar impactors <https://ui.adsabs.harvard.edu/abs/2023E%26PSL.60217963X/abstract>

Topographic Diffusion Revisited: Small Crater Lifetime on the Moon and Implications for Volatile Exploration <https://ui.adsabs.harvard.edu/abs/2022JGRE..12707510F/abstract>

The sizes of the other minor bodies; 2020 PN1, 2020 PP1, and 2020 KZ2 are estimated to be 10-50 m in diameter (assuming albedo 0.20-0.04, see [1–3]). Using the formula from Melosh (2020) and the ejection speeds considered in the paper, ejecta of 10-50 m in diameter are produced from craters of size range 10 - 120 km. This is illustrated below in Figure R2. Smaller scale impacts as those in the references provided by the reviewer produce crater sizes of a few meters or at most 100 meters, too small to explain the size of these bodies.

REVIEWER 2:

1) There is a simple equation in Melosh (2020) allowing to estimate the size of the largest fragment ejected from an impact crater. It may be better to use it (and the result does not contradict your estimates). It is obvious from this equation that the fragment size correlates negatively with an impact velocity - there is no need to mention exceptionally high impact velocities on the Moon.

The formula by Melosh does agree with our estimate as can be appreciated in Figure R2, where the lines in *purple* and *green* indicate the expected diameter of the largest lunar ejecta as a function of the crater diameter for ejecta speeds 2.4 km/s and 6.0 km/s we considered.

2) The statement "ejected debris may achieve speeds up to 6 km/s" (line 546) is not quite correct - the speed could be as high as an impact velocity, but the fraction of this super-high speed ejecta is negligible. Also it is worth to mention that the total fraction of solid high velocity ejecta is relatively small in comparison with molten high-velocity ejecta.

We agree and have revised that statement to clarify the point about the very high speed ejecta, and added two relevant references.

The total number of > 30-km-diameter craters of Copernican period could be estimated from the Neukum's production function (also a very easy exercise) with the reference to Neukum et al. (2001) paper. It is indeed 44.

All the craters of > 30-km-diameter of Copernican period remain observable and countable to the present day. Our statement is based on the actual observations, so we consider redundant to include the estimate from Neukum's production function which is a model based on the observations plus some assumptions, and would be most relevant for estimating the numbers of craters that may have become unrecognizable/uncountable over time.

REVIEWER 3:

This paper explores the hypothesis that Kamo was originated from the fragments from a meteoroidal impact with the lunar surface, with intensive numerical simulations. The results are exciting and supports this hypothesis to a very good level. It is a very interesting and original research to the community and the wider field. Some main concerns include:

1) how is the reliability of the numerical simulations, like how to validate your numerical simulations, maybe even partially;

We used the IAS15 N-body orbit integrator in the REBOUND software package [4], a standard integrator for solar system dynamics applications. We compared the output of this integrator with the ephemeris from JPL Horizons. After a 500 years propagation (starting at January 1 2000) the fractional difference between JPL's and our computation of Kamo'oalewa's semi-major axis is $\sim 3 \times 10^{-6}$. A stringent test is that Kamo'oalewa's HS-QS transitions found in the ephemeris are faithfully reproduced by our simulation.

2) could you please discuss the extent of the sensitivity? For instance, if the initial condition is perturbed for about 1%, how much the error of the orbit will be at 5000 years time? In addition, does the theoretical assumption have an influence on the numerical conclusion? The statistical analysis is appropriate but its validity needs to be discussed more. I think the level of detail provided are good enough to reproduce the work.

Individual cases are very sensitive to initial conditions, as expected for a chaotic system like this one. As an example, Figure R3 shows the semimajor axis evolution of two particles, one launched with a speed of 3.5 km/s from the lunar trailing-side, and another with the same launch site and launch velocity direction but a 1% higher speed. These two particles diverge significantly after just a couple of months, although they remain in the orbital space of near-Earth asteroids during the considered time span.

We also made a comparison between the statistical outcomes over 5000 years of evolution of two sets of 100 test particles, one with a launch speed of 3.5 km/s and the other with a 1% greater speed; the launch directions were chosen uniformly as described in the manuscript;

Fig. R3

all other conditions were the same for the two sets. Figure R4 shows the evolution of the two sets of particles in the (a, e) plane. We observe that they evolve chaotically but overall quite similarly.

For the first set, we found 29 particles reaching a HS orbit, and 1 reaching a HS–QS orbit. For the second set, 26 reached a HS orbit, and 1 reached a HS–QS orbit. A similar numerical experiment was repeated for a launch speed of 2.7 km/s, a regime where collisions (with Earth and Moon) are more frequent. For the first set of particles we found 21 of them reaching a HS orbit, and 1 reaching a HS–QS orbit. For the second set, we found 22 reaching HS orbits and 0 reaching HS–QS orbits. Given the rarity of co-orbital outcomes in this system, we consider that this provides validation of the consistency for the statistics of outcomes.

Some minor comments given here:

1) Page 6 line 242, Please justify why the accuracy is set to be 10^{-9} , rather than other values such as 10^{-12} ?

IAS15 is an adaptive step-size integrator. This accuracy parameter is an internal parameter which controls the step size of the integration. A full discussion can be found in [4], where it is demonstrated that machine precision accuracy (16 significant decimal digits) of the orbit propagations is reached using $\epsilon \approx 10^{-7}$, but a more conservative choice of 10^{-9} is set as default. We have clarified the text to prevent confusion.

2) Page 6 line 254, Why you select 22th Dec 2003 for the initial conditions? Please explain what is Copernican principle? Please add reference

The epoch of the initial conditions was chosen arbitrarily from within the JPL ephemeris time range available for Kamo'oalewa. (It is the lead author's sister's birthday.) The Copernican principle is a well-known principle that states that we are not privileged observers, not in time nor in space [5, 6]. We added these references to the manuscript.

Fig. R4

3) Page 6 line 260, please explain the definition of θ_1 and θ_2

These were explained in Section 3, with the aid of Fig. 2; in the revised paper we improved the wording in Section 3 (please see 3rd paragraph, lines 298-301 in the revised manuscript). θ_1 is the angle between the line that joins the center of the Moon to the center of the Earth and the line that joins the center of the Moon to the launch point on the lunar surface, θ_2 is the angle between the launch velocity vector and the local normal vector at the launch point on the lunar surface.

4) Fig.7 Is the inclination evolution of Kamo from pure numerical simulations? Are there any observation data that can fit part of the evolution curve?

Kamo'oailewa's inclination in Fig. 7 is from JPL's ephemeris. JPL's ephemeris for Kamo'oailewa's orbital parameters for the time range 1599–2500 CE is based on observational data (including 310 observations of Kamo'oailewa from 2004-03-17 to 2021-05-13). The longer timescale evolutions discussed in our work are based on the orbit propagation with a Solar System model and the IAS15 integrator as explained in the manuscript. These simulations closely reproduce the JPL ephemeris for Kamo'oailewa.

Other revisions

- We revised section 2 (theoretical estimates) for greater clarity and accuracy.
- We revised the abstract slightly to highlight the rarity of Kamo'oailewa-like outcomes for lunar ejecta.
- We changed the notation for the launching speed of ejected particles, now it is denoted as v_L .

References

- [1] Small Body Database Lookup, 2020 PN1. https://ssd.jpl.nasa.gov/tools/sbdb_lookup.html#/?sstr=54050997 Accessed 2023-06-30

- [2] Small Body Database Lookup, 2020 PP1. https://ssd.jpl.nasa.gov/tools/sbdb_lookup.html#/?sstr=2020%20PP1 Accessed 2023-06-30
- [3] Small Body Database Lookup, 2020 KZ2. https://ssd.jpl.nasa.gov/tools/sbdb_lookup.html#/?sstr=2020%20KZ2 Accessed 2023-06-30
- [4] Rein, H., Spiegel, D.S.: IAS15: a fast, adaptive, high-order integrator for gravitational dynamics, accurate to machine precision over a billion orbits. *Mon. Not. R. Astron. Soc.* **446**, 1424–1437 (2014)
- [5] Peacock, J.A.: *Cosmological Physics*, (1999)
- [6] Gott, I. J. R.: Implications of the Copernican principle for our future prospects. *Nature* **363**(6427), 315–319 (1993)

1st Sep 23

Dear Mr Castro-Cisneros,

Your manuscript titled "Orbital pathways for a Lunar-Ejecta Origin of the Near-Earth Asteroid Kamo`oalewa" has now been seen by our reviewers, whose comments appear below. In light of their advice we are delighted to say that we are happy, in principle, to publish a suitably revised version in Communications Earth & Environment under the open access CC BY license (Creative Commons Attribution v4.0 International License).

We therefore invite you to revise your paper one last time to address the remaining concerns of our reviewers. At the same time we ask that you edit your manuscript to comply with our format requirements and to maximise the accessibility and therefore the impact of your work.

EDITORIAL REQUESTS:

*****Please take care to match our formatting and policy requirements. We will check revised manuscript and return manuscripts that do not comply. Such requests will lead to delays. *****

SUBMISSION INFORMATION:

OPEN ACCESS:

Communications Earth & Environment is a fully open access journal. Articles are made freely accessible on publication under a [CC BY license](http://creativecommons.org/licenses/by/4.0) (Creative Commons Attribution 4.0 International License). This license allows maximum dissemination and re-use of open access materials and is preferred by many research funding bodies.

For further information about article processing charges, open access funding, and advice and support from Nature Research, please visit <https://www.nature.com/commsenv/article-processing-charges>

At acceptance, you will be provided with instructions for completing this CC BY license on behalf of

all authors. This grants us the necessary permissions to publish your paper. Additionally, you will be asked to declare that all required third party permissions have been obtained, and to provide billing information in order to pay the article-processing charge (APC).

[link redacted]

Best regards,

Joe Aslin

Senior Editor,
Communications Earth & Environment
<https://www.nature.com/commsenv/>
Twitter: @CommsEarth

REVIEWERS' COMMENTS:

Reviewer #1 (Remarks to the Author):

Journal: Nature Communications Earth & Environment

Manuscript #: COMMSENV-23-0603A

Title: Orbital pathways for a Lunar-Ejecta Origin of the Near-Earth Asteroid Kamo'oailewa

Authors: J. D. Castro-Cisneros, R. Malhotra, and A. J. Rosengren

The authors have satisfactorily addressed all the points raised in my original report and I consider that the revised manuscript is almost ready for publication. However, a new orbital distribution model for NEOs has recently been released and published here:

NEOMOD: A New Orbital Distribution Model for Near-Earth Objects
<https://ui.adsabs.harvard.edu/abs/2023AJ....166...55N/abstract>

and I would like to ask the authors to investigate if this new model can reproduce the current orbital status of 469219 Kamo'oailewa (2016 HO3). The authors of this new work provide a link to the source code and

auxiliary files of the model so it should not be difficult to perform this final check with the materials to be found here:

https://www.boulder.swri.edu/~davidn/NEOMOD_Simulator

Unless the result of this additional check comes into conflict with what was obtained with NEOPOP, I do not need to see the revised version of this manuscript again.

Reviewer #2 (Remarks to the Author):

The authors accepted all my comments (minor) and the MS can be published as it is.

Reviewer #3 (Remarks to the Author):

My comments are well addressed.

Manuscript COMMSENV-23-0603
Response to Reviewers (2nd Round)

Authors' responses are in bold font.

REVIEWER 1

The authors have satisfactorily addressed all the points raised in my original report and I consider that the revised manuscript is almost ready for publication. However, a new orbital distribution model for NEOs has recently been released and published here:

NEOMOD: A New Orbital Distribution Model for Near-Earth Objects <https://ui.adsabs.harvard.edu/abs/2023AJ....166...55N/abstract>.

and I would like to ask the authors to investigate if this new model can reproduce the current orbital status of 469219 Kamo‘oalewa (2016 HO3). The authors of this new work provide a link to the source code and auxiliary files of the model so it should not be difficult to perform this final check with the materials to be found here:

https://www.boulder.swri.edu/~#x223C;davidn/NEOMOD_Simulator.

Unless the result of this additional check comes into conflict with what was obtained with NEOPOP, I do not need to see the revised version of this manuscript again.

Based on a realization of this new orbital distribution model provided by David Nesvorny, we found again zero objects with orbital parameters among the values reached by Kamo‘oalewa along the 900 years provided by JPL Horizons. We have added a paragraph in the Introduction section indicating that such a check of the most recent NEAs model was made by the authors.

REVIEWER 2:

The authors accepted all my comments (minor) and the MS can be published as it is.

REVIEWER 3:

My comments are well addressed.